TRANSPARENT OPEN
PROCESS ACCESS

# Evidence for rate-dependent filtering of global extrinsic noise by biochemical reactions in mammalian cells

Jiegen Wu[1,2,3] (iD), Xu Han[1,2], Haotian Zhai[1], Tingyu Yang[1] & Yihan Lin[1,2,*] (iD)

## Abstract

Recent studies have revealed that global extrinsic noise arising from stochasticity in the intracellular biochemical environment plays a critical role in heterogeneous cell physiologies. However, it remains largely unclear how such extrinsic noise dynamically influences downstream reactions and whether it could be neutralized by cellular reactions. Here, using fluorescent protein (FP) maturation as a model biochemical reaction, we explored how cellular reactions might combat global extrinsic noise in mammalian cells. We developed a novel single-cell assay to systematically quantify the maturation rate and the associated noise for over a dozen FPs. By exploiting the variation in the maturation rate for different FPs, we inferred that global extrinsic noise could be temporally filtered by maturation reactions, and as a result, the noise levels for slow-maturing FPs are lower compared to fast-maturing FPs. This mechanism is validated by directly perturbing the maturation rates of specific FPs and measuring the resulting noise levels. Together, our results revealed a potentially general principle governing extrinsic noise propagation, where timescale separation allows cellular reactions to cope with dynamic global extrinsic noise.

**Keywords** biological noise; chromophore maturation; fluorescent protein; global extrinsic noise

**Subject Categories** Computational Biology; Translation & Protein Quality

**Mol Syst Biol. (2020) e9335**

## Introduction

Stochastic fluctuations or noise are inevitable for reactions occurring inside the cell (McAdams & Arkin, 1997; Elowitz *et al*, 2002; Paulsson, 2004; Raser & O'Shea, 2005; Raj & van Oudenaarden, 2008; Eldar & Elowitz, 2010). A key reason is that for some cellular reactions (Fig 1A), the molecular species involved often have low copy numbers and are subject to random birth and death processes,

leading to Poisson-like fluctuations (Swain *et al*, 2002; Paulsson, 2004). This source of noise represents a type of noise that is intrinsic to the reaction of interest and can propagate in biological networks (Fig 1B, left). In addition to intrinsic noise, cellular reactions are also subject to extrinsic noise—fluctuations that are extrinsic to the reaction of interest (Swain *et al*, 2002; Paulsson, 2004; Shahrezaei *et al*, 2008). While some sources of extrinsic noise are specific to certain reactions, other extrinsic noises are global and may affect many reactions (Pedraza & van Oudenaarden, 2005; Raser & O'Shea, 2005). Both intrinsic and extrinsic noises have been characterized in many biological processes, especially in gene regulation, and can play important roles in phenotypic heterogeneities at the cellular or the organismal level (Raser & O'Shea, 2005; Raj & van Oudenaarden, 2008; Eldar & Elowitz, 2010).

Recently, a growing interest has been drawn to a type of global extrinsic noise arising from fluctuations in the intracellular biochemical environment (Slavov *et al*, 2011; Labhsetwar *et al*, 2013; Kiviet *et al*, 2014; Kotte *et al*, 2014; Xiao *et al*, 2016; Ahn *et al*, 2017; Hung *et al*, 2017; Papagiannakis *et al*, 2017; Wehrens *et al*, 2018; Yugi & Kuroda, 2018; Zhang *et al*, 2018; Evers *et al*, 2019; Tonn *et al*, 2019), often reflected as fluctuating metabolic state or metabolite concentration (Yang *et al*, 2008; Ahn *et al*, 2017; Hung *et al*, 2017; Zhang *et al*, 2018; Evers *et al*, 2019) (Fig 1B, right). Importantly, such noise typically occurs at a timescale shorter than a cell cycle and can arise from the stochastic expression or post-translational modification of the enzymes involved in metabolic reactions (Wehrens *et al*, 2018; Yugi & Kuroda, 2018). Since metabolites can bind to many proteins to regulate their activities (Li *et al*, 2010), a fluctuating metabolite concentration could lead to global protein activity fluctuations, which may affect reactions they catalyze. Several experiments and models have implicated that such global extrinsic noise can result in non-genetic cell-to-cell variability in physiological states, including cell growth and drug resistance (Kiviet *et al*, 2014; Kotte *et al*, 2014; Charlebois *et al*, 2018; Rosenthal *et al*, 2018; Thomas *et al*, 2018; Farquhar *et al*, 2019; Kheir *et al*, 2019; Tonn *et al*, 2019; Xiao *et al*, 2019). However, it has been experimentally challenging to analyze the effects of this source of noise, and it is unclear

1 Center for Quantitative Biology and Peking-Tsinghua Joint Center for Life Sciences, Academy for Advanced Interdisciplinary Studies, Peking University, Beijing, China
2 The MOE Key Laboratory of Cell Proliferation and Differentiation, School of Life Sciences, Peking University, Beijing, China
3 Tsinghua-Peking Joint Center for Life Sciences, Tsinghua University, Beijing, China
   *Corresponding author. Tel: +86-10-62758034; E-mail: yihan.lin@pku.edu.cn

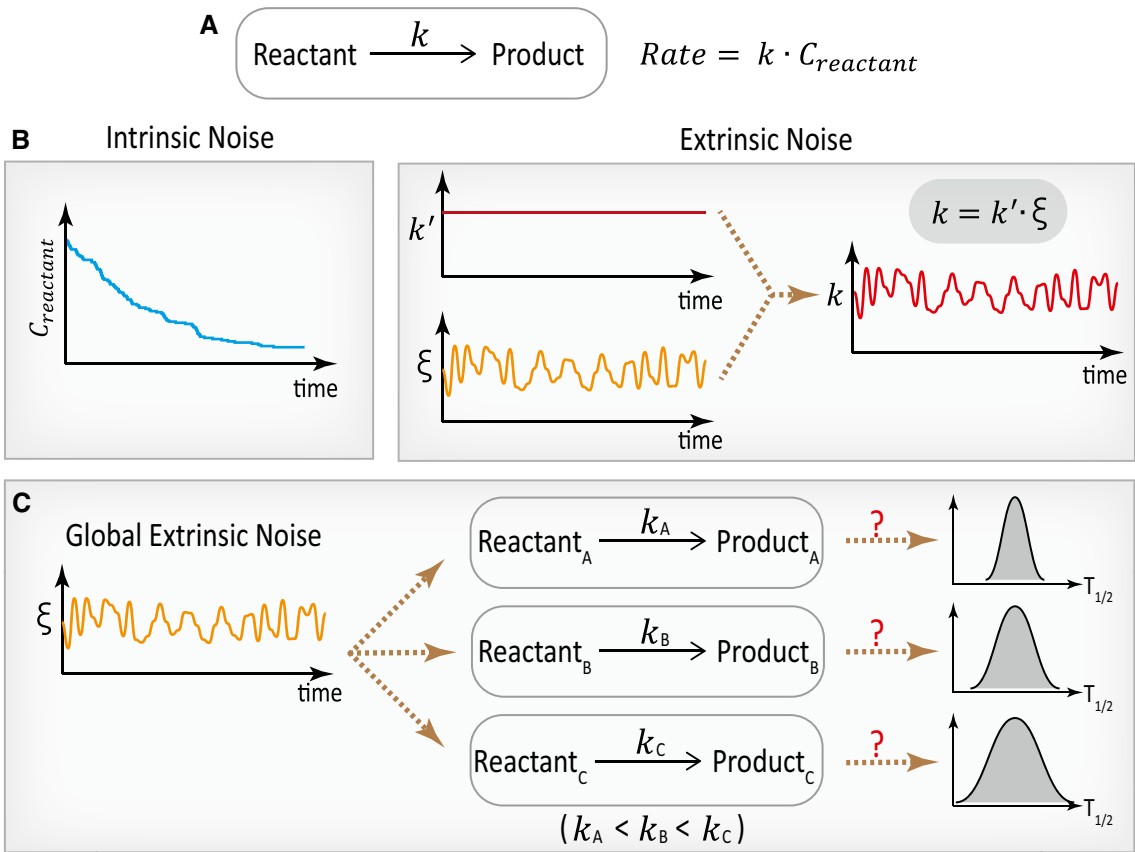

**Figure 1. Biochemical reactions inside the cell are subjected to both intrinsic noise and extrinsic noise.**

A Schematic of an example first-order reaction with a rate constant of $k$. The expression for the reaction rate is shown on the right, where $C_{reactant}$ denotes the cellular concentration of the reactant.

B Schematic representations for intrinsic noise (left) and extrinsic noise (right). Intrinsic noise arises from the low copy number nature for some intracellular molecules. The schematic on the left shows the fluctuations of reactant concentration along an exponential decay curve. The schematic on the right illustrates the effect of extrinsic noise on the rate constant $k$, resulting in a dynamically fluctuating rate constant. Extrinsic noise can come from the fluctuations in the upstream components.

C Global extrinsic noise can affect many biochemical reactions inside the cell. It has remained largely unclear how downstream reactions might combat with dynamic global extrinsic noise and whether the timescale of the reaction plays a role in affecting the cell-to-cell variability in the reaction.

whether different downstream reactions are equally affected by such dynamic fluctuations (Fig 1C).

To address these issues, we utilized FP maturation reaction as a model biochemical reaction inside the cell and investigated how extrinsic noise propagates from upstream biochemical environment to downstream maturation reaction. The rationale for focusing on the maturation reaction of FPs is at least twofold. Firstly, FP maturation is mostly orthogonal to other cellular reactions and is influenced by the metabolic environment of the cell, such as the NAD(P)H level (Elsliger *et al*, 1999; Zhang *et al*, 2006; Ganini *et al*, 2017), making it an ideal reaction for studying how it is affected by environmental fluctuations. Secondly, chromophore maturation reaction typically occurs at the timescale of minutes to hours, depending on the type of the FP (Balleza *et al*, 2017; Lambert, 2019). In contrast, the timescale of fluctuations in intracellular biochemical environment likely ranges from seconds to minutes, estimated from previous live-cell measurements of metabolic state dynamics using real-time biosensors for metabolites, including NADH, ATP, and lactate (San *et al*, 2013; Hung *et al*, 2017; Tao *et al*, 2017; Depaoli *et al*,

2018). The actual timescale is likely much shorter as metabolite biosensors often have low temporal sensitivity. Thus, this global extrinsic noise presumably occurs at a faster timescale compared to the downstream maturation reaction, allowing studying the role of timescale separation in extrinsic noise propagation.

To study how extrinsic noise is propagated to FP maturation reaction, we first developed an assay that decouples protein expression, and chromophore maturation into two separate signals, enabling us to systematically analyze maturation rates and the associated noise levels in individual mammalian cells. Using this assay, we quantified the maturation rates of 14 commonly used FPs and found that the timescale of the reaction spans from ~10 to ~140 min. Based on these single-cell data, we computed the noise level for the maturation reaction of each FP and identified a surprising correlation between the noise level and the rate of maturation, where the slow-maturing ones have lower levels of noise. We next provided *in silico* and *in vivo* evidences supporting a mechanism in which the global extrinsic noise is temporally filtered in a rate-dependent manner, leading to reduced noise levels for the slower

reactions. Thus, the timescale of the downstream reaction determines the degree of stochasticity inherited from its biochemical environment. Furthermore, since this is the first systematic study, to our knowledge, on FP maturation in mammalian systems, we carried out in-depth characterizations regarding the susceptibility of the maturation kinetics to various parameters and identified limitations when using FPs to measure dynamic and stochastic processes in mammalian cells. Together, these results not only offer new knowledge regarding FPs in mammalian cells, but also uncover a principle governing extrinsic noise transmission in stochastic biochemical environment, which could be general for diverse cellular reactions.

# Results

## A rationally designed assay for quantifying FP maturation rate in individual mammalian cells

The process of FP chromophore maturation involves multiple chemical reaction steps and is typically described as a single first-order reaction, whose rate constant determines the timescale of the maturation reaction (Reid & Flynn, 1997; Zhang *et al*, 2006; Iizuka *et al*, 2011). Many efforts have been devoted to characterizing FP maturation rates using *in vitro* assays (Tsien, 1998; Shaner *et al*, 2005; Day & Davidson, 2009; Iizuka *et al*, 2011). Systematic *in vivo* studies have been carried out mostly in bacterial (Hebisch *et al*, 2013; Balleza *et al*, 2017) or yeast cells (Gordon *et al*, 2007; Shashkova *et al*, 2018). In these assays, protein synthesis inhibitors are used to ensure that only already synthesized proteins undergo the maturation reaction. However, such proteome perturbations may alter cellular physiological conditions and affect chromophore maturation.

To accurately characterize the rate of FP maturation reaction in cultured mammalian cells without using protein translation inhibitors, we sought to decouple the two temporally coupled reactions, protein production and chromophore maturation, into two spectrally distinct fluorescence signals. To achieve this, we modified a translocation-based assay (Aymoz *et al*, 2016) to quantify the production and maturation of FP molecules independently, in which the amount of expressed FP molecules is measured by the cytoplasmic-to-nuclear translocation of a separate and constitutively expressed FP (Fig 2A). In so doing, we can quantify the amount of expressed FP molecules as well as the amount of matured FP molecules separately over time, from which the FP maturation rate can be determined.

More specifically, in this assay, a constitutively expressed and cytoplasmic localized FP (i.e., the constitutive FP, which has no nuclear localization sequence or NLS) is used for quantifying the expression of the NLS-containing target FP, which is under doxycycline-inducible control (i.e., the FP of interest) (Figs 2A and EV1A–C). The two spectrally separated FPs, the constitutive FP and the target FP, are each fused with heterospecific SynZip domains (Reinke *et al*, 2010; Aymoz *et al*, 2016). Upon the induced expression of the target FP (by adding doxycycline), the constitutive FP molecules form dimers with the unmatured target FP molecules, which are then shuffled into the nucleus due to the NLS on the target FP molecules. As a control, we ensured that the addition of doxycycline does not affect the production or

localization of the constitutive FP molecules (Fig EV1D). Therefore, the nuclear accumulation of the constitutive FP fluorescence signal can be used for quantifying the expression level of the target FP, whose maturation can be separately quantified by the increase of its own nuclear fluorescence signal. Take mKate2 for example, this cell line contains constitutively expressed Citrine, inducible mKate2, and H2B-mTurquoise2. During the experiment, after the addition of doxycycline, all three fluorescence signals were recorded. The resulting time-lapse images of H2B-mTurquoise2 were processed with MATLAB codes to extract the masks of the cell nucleus, from which we can track the movements of individual cells over time (Fig 2B, Movie EV1). Tracks of nuclear masks were then used to quantify the changes in the nuclear localization of Citrine, together with the changes in the nuclear mKate2 signals (Fig EV1B). These data allowed us to determine mKate2 protein production rate as well as its maturation rate on a cell-by-cell basis (Fig 2C). Therefore, by effectively decoupling the two temporally coupled processes into distinct signals, we can determine the maturation rate in individual cells by fitting these two temporal signals with simple kinetic models (see Materials and Methods).

Since the maturation reaction occurs inside the nucleus in our assay, we sought to determine whether FP matures at a similar rate inside the nucleus as in the cytoplasm. To perform this analysis, we used a bidirectional promoter that drives the expression of two identical FPs but with different subcellular localizations (Fig EV2A). Cells were induced to express these two FPs at the same time, and the cytoplasmic and nuclear fluorescence levels were measured at different time points post-induction. By analyzing a large population of cells with high-content microscopy, we found that nuclear and cytoplasmic fluorescence signals are highly correlated throughout the time course of induction, indicating that the rates of FP maturation are similar between the two subcellular locations (Fig EV2B and C, see Materials and Methods).

## Different FPs display variable maturation rates that are robust to diverse parameters

With this assay, we first addressed whether different FPs exhibit variable maturation rates in mammalian cells. We focused on 14 commonly used FPs whose emission spectra span from blue to near-infrared (Thorn, 2017; Lambert, 2019) (Datasets EV1 and EV2). For each FP, we constructed a stable monoclonal Chinese hamster ovary (CHO) cell line that contains the constitutive FP, the target FP, and a third FP for labeling the nucleus (Table EV1, see Materials and Methods).

By analyzing single-cell fluorescence trajectories for each FP (see examples in Figs 2C and EV1B), we obtained the maturation rates for the chosen set of FPs (Figs 2D and EV1E). From these data, we found that the maturation rate is highly variable across the 14 different FPs, with the timescale spanning from ~10 min to ~140 min. This broad range of timescale of the reaction rate will allow us to address how reaction timescale affects noise transmission from upstream fluctuations. From the perspective of FP-based tools, the variability in FP maturation rates presents challenges when comparing quantitative measurements using different FPs, underscoring the importance of maturation rate characterizations. These results also provide a useful resource when choosing FPs to examine temporal

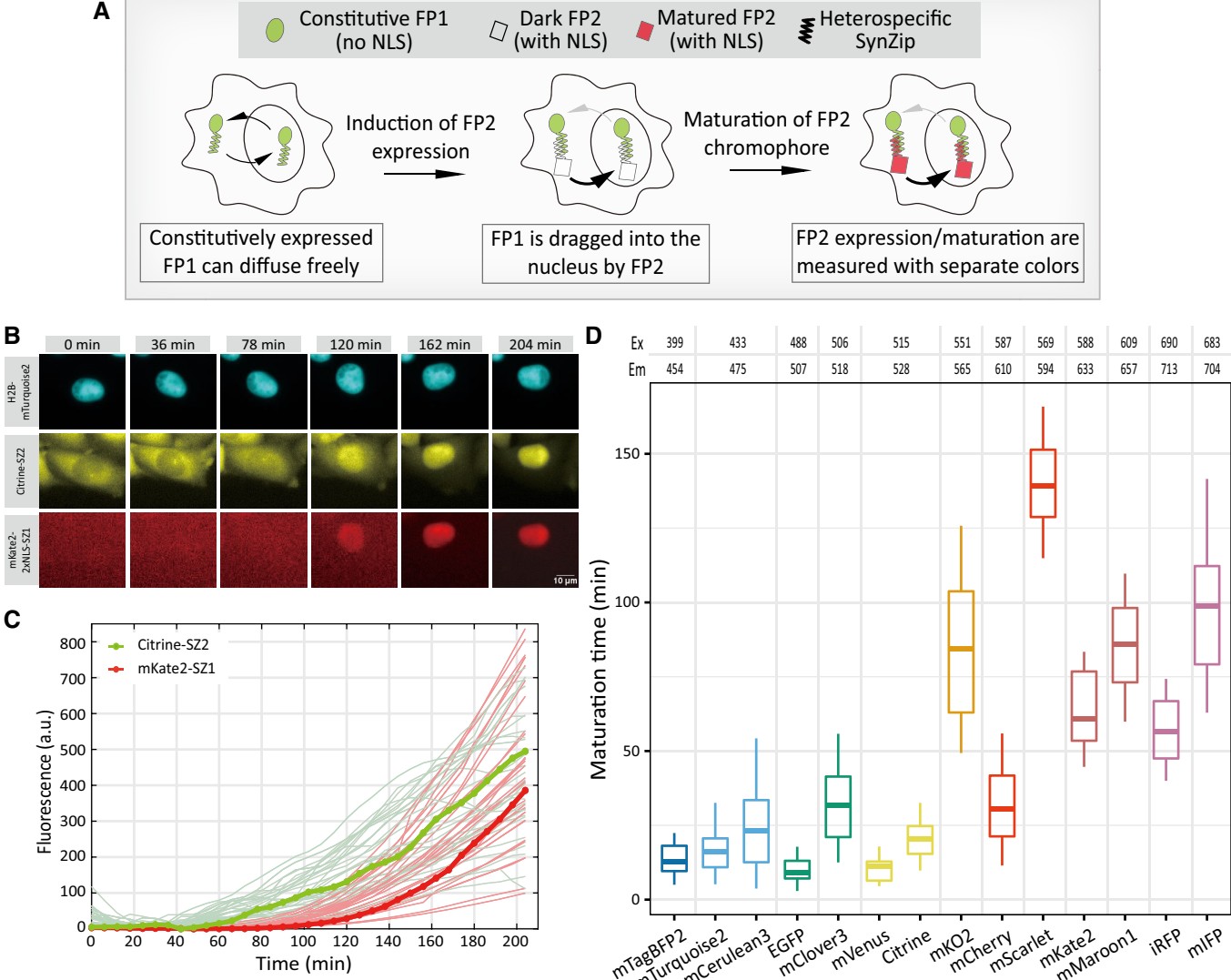

**Figure 2. Systematic characterizations of the maturation rate for 14 different FPs in single mammalian cells.**

A   Schematic of our single-cell assay which decouples FP production and FP maturation into two separate signals in individual mammalian cells. The expression level of FP2 is measured by the nuclear fluorescence intensity signal of FP1, while the maturation of FP2 is measured by its own nuclear fluorescence intensity signal.

B   A representative filmstrip from the microscopy experiments characterizing mKate2 maturation. Fluorescence images for the H2B-mTurquoise2 (for labeling the nucleus, top), Citrine-SZ2 (the constitutive FP or FP1, middle), and mKate2-2xNLS-SZ1 (the inducible FP or FP2, bottom) are shown for indicated time points. The sale bar is 10 μm.

C   Single-cell fluorescence intensity traces for quantifying mKate2 expression and mKate2 maturation. For these data, doxycycline was added at time zero. Traces in bold refer to the cell shown in (B). Traces for other cells (*n* = 30) are shown in light colors.

D   Boxplots showing the maturation times for 14 FPs measured by our assay. FPs are sorted by their spectra properties. Each box ranges from the first quartile to the third quartile of the data values, and the horizontal line inside indicates the median. The upper whisker is drawn up to the largest data value smaller than the third quartile plus 1.5× the interquartile range (IQR), and the lower whisker is drawn up to the lowest data value larger than the first quartile minus 1.5× IQR. The dataset for each FP contains *n* = 16–130 cells. Detailed information regarding the cell lines used can be found in Table EV1.

processes such as gene expression in mammalian cells, as slow-maturing FPs act as a low-pass filter that obscures fast transcriptional activity changes (Nagai *et al*, 2002; Balleza *et al*, 2017). It is of interest to note that red-emitting FPs appear to mature slower in general than blue or green-emitting FPs in CHO cells (Fig 2D), which may arise from the differences in the sequence composition and environment of the FP chromophore peptides (Grigorenko *et al*, 2017).

We next explored the susceptibility of FP maturation rate to a variety of cell state-related parameters. First, by computing a correlation matrix based on the above single-cell data, we found that the maturation rates of individual cells have relatively low correlations with cell state parameters, including initial cell size, level of the constitutive FP, and several others (Fig EV3A and B). Second, by analyzing cells starting from different cell cycle stages (Fig EV3C, see Materials and Methods), we concluded that cells maintain an

environment that is relatively constant throughout the cell cycle for the FP maturation reactions (Fig EV3D, see Materials and Methods). Last, by analyzing cells of different CHO monoclones, we found that maturation does not appear to be influenced by the genetic background (Fig EV3E). Together, these results suggest that the FP maturation process is robust to many cell state-related parameters.

### FP maturation rates exhibit non-genetic heterogeneity

To investigate how noise in the maturation reaction is affected by upstream fluctuations, we first needed to analyze the level of noise in the maturation reaction for different FPs. FPs are often used to study noise in gene expression and cell states, and it is typically assumed that the FP maturation rate is homogenous across isogenic cells. Using the single-cell data, we computed the coefficient of variation as the noise level for the FP maturation time. We found that FP maturation is rather noisy for isogenic CHO cells, and unexpectedly, the level of noise exhibits a correlation with the rate of maturation reaction (Fig 3A). More specifically, the noise in maturation time increases as the maturation time shortens.

Notably, the noise in FP maturation rate shows a weak and statistically insignificant correlation with the noise in FP production rate (Fig 3B). Because FP production rate and FP maturation rate are simultaneously determined in the same single cells, the lack of correlation between the two noises indicates that the noise in FP maturation does not arise from technical sources, which would otherwise affect both measurements in the same manner. More

importantly, this result suggests that FP production and FP maturation are largely decoupled, and the two processes are likely influenced by uncorrelated sources of noise. We note that the effect of protein degradation on the maturation reaction should be minor because the FPs do not contain degradation tags, and the effect of passive dilution should also be minor as we only analyzed cells that did not undergo mitosis.

We next investigated the potential source of such non-genetic heterogeneity. Since the bidirectional promoter-based reporter assay (Fig EV2A) is conceptually analogous to the classical "two-color" assay (Elowitz *et al*, 2002), it would allow us to distinguish intrinsic versus extrinsic sources of noise in the measured fluorescence signals. Using the data from this pseudo "two-color" assay, we found that the strength of the intrinsic noise is considerably smaller compared to that of the extrinsic noise (Fig EV2D), suggesting that extrinsic sources of noise are likely the major contributor to the noise in the maturation reaction.

### Time-averaging of global extrinsic noise may underlie the rate-dependent noise level

Because the noise in the maturation reaction decreases as the reaction rate decreases, it appears that the timescale of the reaction is a key determining factor for the noise level. Timescale has been known to determine the noise of reaction in several other systems, including chemotaxis and developmental pattern formation (Berg & Purcell, 1977; Bialek & Setayeshgar, 2005; Gregor *et al*, 2007;

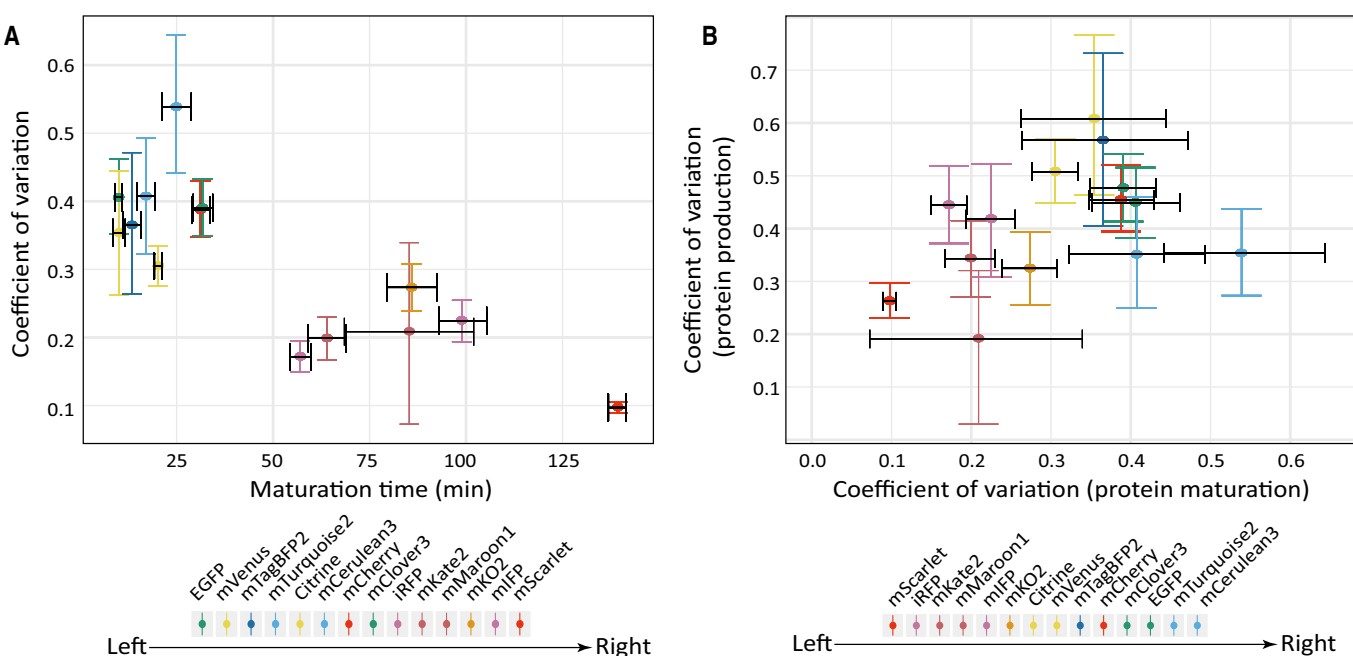

**Figure 3.  The noise level in the maturation reaction rate exhibits a rate-dependent behavior.**

A   Maturation time and the associated cell-to-cell variability for the 14 FPs. Data in Fig 2D were used to compute the coefficient of variation (i.e., the level of noise) for each set of single cells. The noise level shows a statistically significant negative correlation with the maturation time of the FP (Pearson's correlation coefficient: −0.80, *t* = −4.6614, *df* = 12, *P* < 0.001). The dataset for each FP contains *n* = 16–130 cells.

B   Scatter plot of the noise in FP maturation time versus the noise in FP production rate for the 14 FPs. The two noises showed an insignificant correlation (Pearson's correlation coefficient: 0.39, *t* = 1.48, *df* = 12, *P* = 0.16). The dataset for each FP contains *n* = 16–130 cells.

Data information: All error bars indicate 95% confidence intervals of the mean by bootstrap (Materials and Methods).

Tostevin *et al*, 2007). In these systems, in order to achieve accurate responses, cells implement the time-averaging strategy to filter out the fluctuations in upstream signals. In our system, the maturation reaction is assumed to be subjected to the fluctuations in the intracellular biochemical environment such as the NAD(P)H level (Ganini *et al*, 2017) (Fig 4A). A similar time-averaging mechanism could thus take place during FP maturation to account for the observed rate-dependent noise level in the maturation reaction.

To analyze the role of time-averaging in filtering global extrinsic noise, we first illustrated how upstream fluctuations are propagated to the downstream maturation reaction. We used the Gillespie stochastic simulation algorithm to simulate the process of FP maturation in a fluctuating biochemical environment. To reflect the susceptibility of the noise in FP maturation reaction to upstream fluctuations, we imposed a white noise to the first-order maturation

rate constant. In so doing, we observed that the noise in maturation reaction increases as the extrinsic noise level increases (Fig EV4A).

Under the time-averaging scenario, when an FP molecule takes a longer time to mature, the molecule experiences an effectively smaller degree of fluctuations in the intracellular environment due to time-averaging (Fig 4B). In this case, the effective noise in the first-order maturation rate constant would decrease as the maturation rate decreases. By implementing such a time-averaging-based noise filtering mechanism in our stochastic simulations, we indeed observed a dependence of the noise level on the rate of the reaction (Fig 4C, see Materials and Methods). Notably, the simulated results (Fig 4C) qualitatively recapitulated the experimentally observed negative correlation between the noise in FP maturation time and the length of FP maturation time (Fig 3A). Furthermore, at the single FP molecule level, the reaction timescale for each molecule is

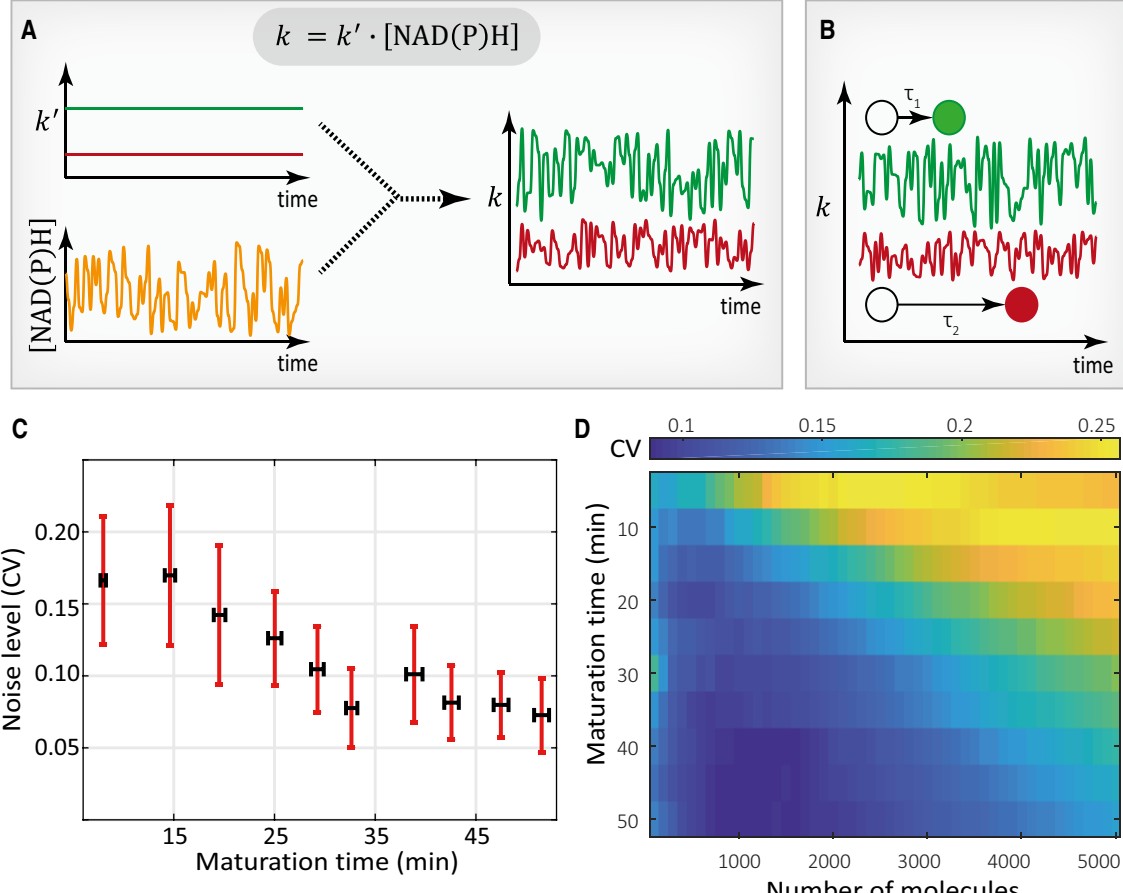

**Figure 4. Time-averaging of global extrinsic noise may account for the rate-dependent noise level in the maturation reaction.**

A   The rate constant of the maturation reaction is subjected to global extrinsic noise (schematic). In this phenomenological model, we assumed that global extrinsic noise arises from fluctuations in the cellular NAD(P)H level (see text for details). The resulting rate constant is stochastic and varies over time. Green line represents the temporally fluctuating rate constant for the FP with a faster maturation rate, while the red line is for the FP with a slower maturation rate.

B   Schematic illustrating the time-averaging of global extrinsic noise. The timescales for the maturation reaction of two FP molecules are illustrated. As a FP molecule takes a longer time to mature (i.e., it has a lower rate constant), it would average the extrinsic noise over a larger time window.

C   Stochastic simulations of the phenomenological model. We incorporated the time-averaging mechanism, and the results recapitulated a negative correlation between FP maturation noise and FP maturation time. Each condition contains 100 cells in the simulation, and each cell contains 2,000 molecules. Error bars indicate 95% confidence intervals of the mean by bootstrap.

D   Stochastic simulations of the model with varying numbers of FP molecules.

not only determined by the mean rate of the reaction but is also inversely proportional to the number of the total FP molecules. Thus, as the molecule copy number increases, the reaction can become more susceptible to the global extrinsic noise (Fig 4D). This result may seem rather counterintuitive because the intrinsic noise level typically decreases as copy number increases, highlighting a unique feature for time-averaging-based filtering of the global extrinsic noise.

### Direct tuning of maturation reaction rate led to altered noise level

Thus far, based on experiments (Fig 3A) and simulations (Fig 4C), we have established that the rate of maturation reaction dictates the level of noise in the reaction through a noise filtering mechanism. Because the experimental support came from the data of different FPs, a potential caveat is that parameters such as the type of FP could play a role. Therefore, we set out to test the proposed mechanism by perturbing the maturation rates of specific FPs and measuring the resulting noise levels. More specifically, for chosen FPs, we asked whether reducing the rate of maturation would promote time-averaging-based noise filtering and thus decrease the noise level (or vice versa).

We first sought to perturb the maturation rate by changing culture conditions. As suggested by previous studies, the critical rate-limiting reaction step during FP maturation is chromophore oxidation (Heim *et al*, 1994; Zhang *et al*, 2006; Iizuka *et al*, 2011), which depends on the dissolved oxygen concentration in the cell culture media (Kaida & Miura, 2012) (Fig 5A). We thus tested whether and how the oxygen level in the cell culture medium affects the FP maturation rate and the associated noise level in maturation. To do so, we pre-treated cell culture media with different concentrations of an oxygen-scavenging enzyme, which created media with a gradient of dissolved oxygen levels (see Materials and Methods). By characterizing single cells in these culture conditions, we found that the FP production rate is not affected by such perturbations, indicating that cells still maintained normal physiological conditions during the experiments (Fig EV4B top). In contrast, reducing the oxygen level significantly slows down the maturation rate as expected (Fig EV4B bottom). Most importantly, slowing down maturation reaction leads to reduced noise levels in the maturation time for two separate FPs (Figs 5B and EV4C–E). Thus, it is evident that the noise level in maturation reaction can be directly tuned by slowing down the reaction, as expected by the proposed mechanism (Fig 4C).

To further test this mechanism, we used an alternative strategy for perturbing the rate of maturation reaction for specific FPs. Because the maturation of phytochrome-based infrared fluorescent proteins such as iRFP and mIFP requires the incorporation of biliverdin as the chromophore, we sought to perturb the maturation rate by varying the amount of biliverdin in the culture medium (Fig 5C) (Shemetov *et al*, 2017). By quantifying the maturation rates of two near-infrared FPs under two different biliverdin concentrations, i.e., 0 and 10 μM, we found that the rates of maturation for both FPs were increased by the addition of biliverdin, and the noise levels were increased accordingly (Figs 5D and EV5). Importantly, the protein production rates were unaffected by the addition of biliverdin (Fig EV5A and B). Furthermore, because the endogenous biliverdin concentration is cell type-dependent (Yu *et al*, 2015), we postulated that the rate of maturation reaction for near-infrared FPs

would differ in different cell types (Fig EV6A). We thus compared their rates in human bone osteosarcoma (U2OS) cells and CHO cells, and found that the maturation rates are consistently slower in CHO cells for both iRFP and mIFP, and intriguingly, the noise levels in the maturation reaction for both FPs are lower in CHO cells (Fig EV6B and C).

Together, using these separate perturbation experiments, we directly tuned the rate of maturation reaction for specific FPs and validated the model prediction, providing further supports for the rate-dependent filtering of dynamic global extrinsic noise.

## Discussion

Thanks to recent processes in metabolite biosensors, fluctuations in intracellular metabolites are increasingly recognized as a key source of extrinsic noise that dynamically influences biochemical reactions inside the cell (Wehrens *et al*, 2018; Yugi & Kuroda, 2018; Evers *et al*, 2019; Tonn *et al*, 2019). In this study, we employed the maturation reaction of FPs as a model biochemical reaction to investigate the principle underlying noise transmission from the intracellular environment to downstream reactions. Through systematic single-cell characterizations of over a dozen FPs, together with direct perturbations of the maturation reaction rate, we showed that the rate of the reaction, or the reaction timescale, plays a critical role in determining the noise level of the reaction, which is likely accomplished through time-averaging of global extrinsic noise. More generally, this mechanism suggests a critical role of timescale separation—the timescales of slower reactions are separated from the timescale of upstream environmental fluctuations, enabling these reactions to combat the fast fluctuating biochemical environment.

Due to the lack of tools to analyze FP maturation reaction in single mammalian cells, we developed a novel single-cell assay to accurately quantify the rate of FP maturation and the associated noise level. In contrast to previous FP maturation assays that rely on protein synthesis inhibitors, our single-cell assay enables independent measurements of the rate of protein production as well as the rate of FP maturation in the same single cells. With such data, we were able to characterize how diverse parameters specifically affect the maturation kinetics without being convoluted by the effects on protein production. Through systematic studies, we found that, while FP maturation is robust to many cell state-related parameters including cell cycle stage and cell size, the maturation of cofactor-dependent FPs is subjected to variations in cofactor concentration in different cell lines. These and other data we presented provide guidelines for choosing FPs to study dynamic and stochastic mammalian processes.

Noise affects various reactions inside the cell, including the maturation reaction of FPs, as we have demonstrated. We showed how FP maturation noise could arise from the stochasticity in the intracellular biochemical environment. Such global extrinsic noise has been shown in recent studies to influence physiological processes such as drug resistance and cell growth (Kiviet *et al*, 2014; Wehrens *et al*, 2018). Critically, we found that time-averaging of global extrinsic noise appears to contribute to the measured rate-dependent noise in the maturation reaction, allowing slower reactions to achieve lower noise levels in their reaction rates. Thus, the relatively fast biochemical fluctuations could be filtered away by reactions that

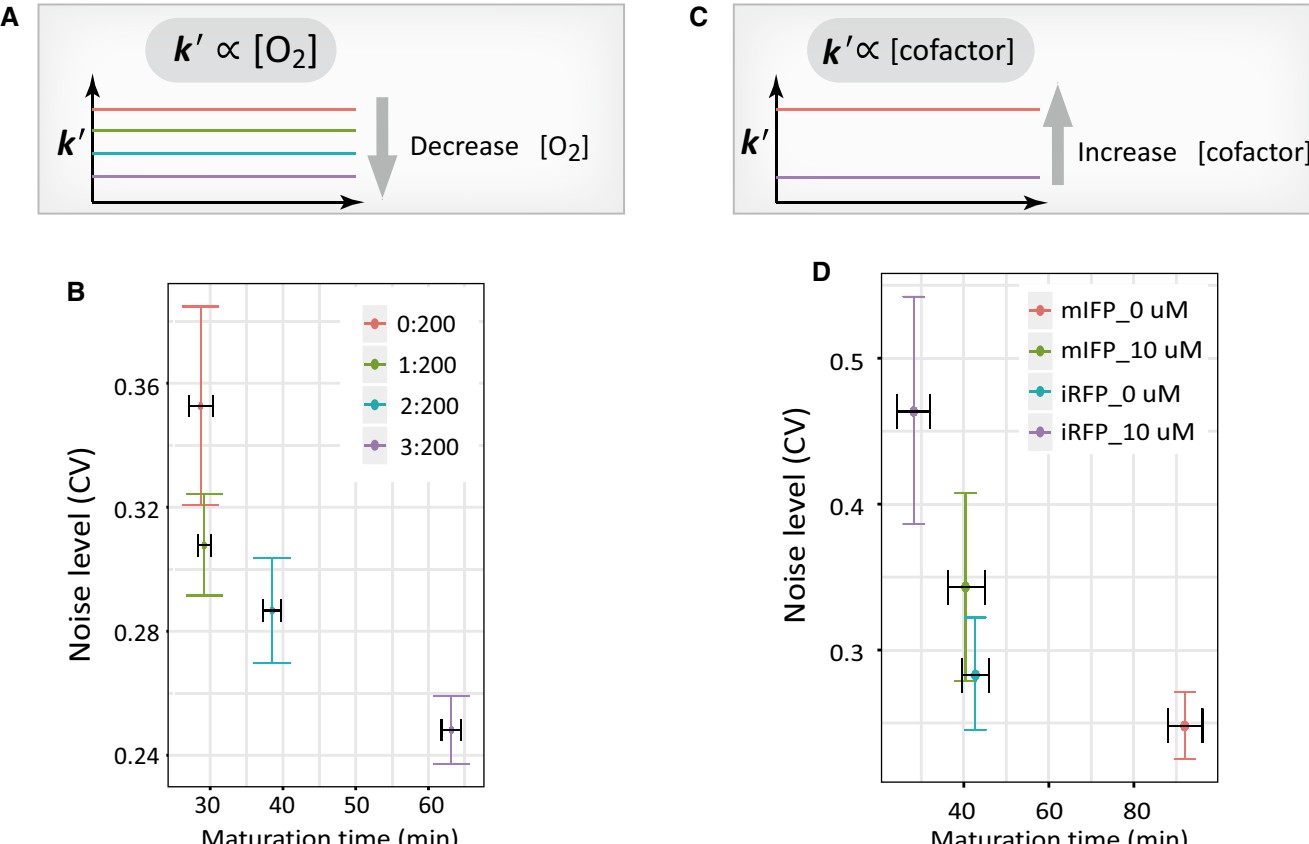

**Figure 5. Further experimental supports for rate-dependent filtering of global extrinsic noise.**

A The maturation rate can be tuned by adjusting cellular oxygen level. The rate constant $k'$ is dependent on the oxygen level as shown by previous studies (Heim *et al*, 1994; Zhang *et al*, 2006; Iizuka *et al*, 2011).

B Reducing the oxygen level in the media by oxygen-scavenging enzyme reduces the maturation rate of mCherry, leading to lower noise levels. In this experiment, culture media were pre-treated with a gradient of an oxygen-scavenging enzyme and the corresponding maturation rates were measured (see Materials and Methods and also Fig EV4B–E). The dataset for each condition (from left to right) contains $n$ = 143, 395, 299, and 529 cells, respectively.

C The maturation rate for cofactor-dependent FPs can be tuned by adjusting cofactor level. The rate constant $k'$ is dependent on the cofactor level as suggested by previous studies (Yu *et al*, 2015).

D Near-infrared FPs exhibited altered maturation rates and the associated noise levels in a biliverdin concentration-dependent manner. Two monoclonal CHO cell lines (mIFP and iRFP) were used, and indicated concentration of biliverdin (or DMSO for 0 μM condition) was added to the culture medium (see also Fig EV5). The dataset for each condition (from left to right) contains $n$ = 40, 39, 56, and 116 cells, respectively.

Data information: All error bars indicate 95% confidence intervals of the mean by bootstrap.

are sufficiently slow, preventing further the propagation of global extrinsic noise. Time-averaging (or temporal averaging/integration) is a widely implemented mechanism for noise reduction in biological systems. For gene expression, time-averaging occurs at different levels, e.g., the integration of noisy mRNA copy numbers into the relatively stable protein counts (Paulsson, 2004; Eldar & Elowitz, 2010). During bacterial chemotaxis, the rapid and stochastic light-receptor binding events are temporally averaged to achieve accurate responses (Berg & Purcell, 1977). During developmental pattern formation, upstream regulatory signals are temporally and even spatially averaged to gain accurate patterns (Gregor *et al*, 2007; Tostevin *et al*, 2007). More generally, time-averaging can be implemented by open-loop systems requiring no feedback mechanism. In contrast, feedback control mechanisms have been well known to reduce noise and confer robustness for several biological systems (Barkai & Leibler, 1997; Arkin *et al*, 1998; Becskei & Serrano, 2000; Nevozhay *et al*, 2009).

Because of the intricate nature of the cellular biochemistry, we do not have a mechanistic picture of how the chromophore maturation reaction is dynamically coupled to the biochemical environment; yet a recent study has revealed the key role of NAD(P)H in FP chromophore maturation (Ganini *et al*, 2017), whose intracellular concentration displays dynamics at the timescale of seconds (Hung *et al*, 2017; Tao *et al*, 2017). Together with our results that FP maturation rate is highly susceptible to environmental oxygen level, it is interesting to speculate that the extrinsic noise source for FP maturation reaction may arise from the intracellular redox environment, whose stochastic fluctuations could potentially influence diverse metabolic and signaling enzymes and contribute to non-genetic heterogeneity of cell state. Since many metabolites display rapid temporal dynamics in the cell (Ahn *et al*, 2017; Wehrens *et al*, 2018; Yugi & Kuroda, 2018), it would be of interest to examine how global extrinsic noises originated by these dynamics propagate to regulate the heterogeneity in cell physiology (Thomas *et al*, 2018).

With the advance in biosensor technology, we may soon be able to search for more general principles governing extrinsic noise propagation through real-time monitoring of the dynamics of key metabolites and downstream reactions simultaneously in the same cells.

# Materials and Methods

### Cell culture

CHO cells were cultured in RPMI 1640 media (Gibco) with 10% FBS (Gibco), supplemented with 1% Pen-Strep and 1× glutamine (Gibco). U2OS cells were cultured in DMEM media (Gibco) with 10% FBS (Gibco), supplemented with 1% Pen-Strep and 1× glutamine (Gibco). Culture media were replaced once every day, and cells were passaged once every 3 days. All cultures were kept under 5% $CO_2$ and 37°C temperature.

### Plasmid construction

All fluorescence protein DNA sequences (Dataset EV2) were synthesized or PCR assembled, and checked by Sanger sequencing. All plasmids were constructed by routine molecular cloning protocols including ligation and Gibson assembly. Plasmids were replicated in DH5α cells using standard protocols. All transgenes in this work were cloned into the plasmids from the PiggyBac transposon system (System Biosciences).

### Cell transfection

Cells were plated into wells in a 24-well plate $\sim 12$ h prior to transfection. And the plating density was controlled such that the culture would arrive at a certain confluency at the time of transfection (60% for CHO cells and 90% for U2OS cells). Plasmids were transfected into cells using Lipofectamine LTX with PLUS™ Reagent (Invitrogen) using standard transfection protocols. For transfecting a well, we used 0.5 μl Plus reagent, 2.0 μl LTX reagent, and 0.8 μg DNA.

To ensure high expression levels for constitutively expressed fluorescent proteins, the mass ratio is 3:1 between the plasmid containing the constitutively expressed fluorescent protein (FP1) and the plasmid containing the inducible fluorescent proteins (FP2), i.e., 300 and 100 ng in a well, respectively. The rest of the 400 ng plasmids contained other components including 150 ng of PiggyBac transposon plasmid, 100 ng of the rtTA (reverse tetracycline-controlled trans-activator) plasmid, and 150 ng of the plasmid for labeling the nucleus using H2B.

### Monoclonal cell line construction

Monoclonal cell lines were obtained from single cells deposited into 96-well plates using fluorescence-activated cell sorting (FACS). Prior to sorting, the inducible fluorescent protein was induced by adding 1 μg/ml doxycycline (Clontech) into the culture media for 12 h. Triple-positive (the constitutive FP or FP1, the inducible FP or FP2, and the nuclear labeling H2B-FP) cells were gated and deposited. The resulted single cells were cultured and expanded in doxycycline-free media.

### Time-lapse fluorescence microscopy

Cells were seeded into wells in a 24-well glass-bottom plate (Eppendorf) several hours ($\sim 36$ h for CHO cells and $\sim 12$ h for U2OS cells) prior to imaging to ensure certain confluency level at the time of imaging ($\sim 90\%$ for CHO cells and $\sim 60\%$ for U2OS cells). To facilitate accurate cell segmentation, we typically added wild-type cells that have no fluorescence into cells of interest at a ratio of about 3:1 (WT cells versus fluorescent cells). By doing so, the segmentation algorithm can achieve a better identification and separation of fluorescent cells.

Time-lapse microscopy was performed on an automated microscope (Nikon Ti-E) using a Plan Apo Lambda 40× objective. The microscope is equipped with a white-light LED (Lumencor SOLA) for fluorescence excitation, standard fluorescence filter sets (Chroma and Semrock) for fluorescence imaging, an automated sample stage for moving between imaging positions, and a scientific CMOS camera (Hamamatsu ORCA-Flash4.0V2) for recording images. The glass-bottom culture plate was maintained in a home-made environmental chamber set at 37°C temperature, and the cells were under a continuous airflow that was pre-humidified and contained 5% $CO_2$. Multi-color images were periodically and automatically acquired using the open-source Micro-Manager program with different frame rates for different types of experiments (1 per 6 min for maturation kinetics measurements unless otherwise specified).

The experimental time course was illustrated in the upper panel of Fig EV1C. More specifically, during these experiments the inducer doxycycline was added to the cell culture in order to induce the expression of the FP of interest. For maturation kinetics measurements, culture media containing 1 μg/ml of doxycycline were added to the wells at time zero, after which images were acquired continuously for 6 h.

### Cell segmentation and tracking

Images were loaded into MATLAB (MathWorks) and analyzed by using a published mammalian cell tracking and segmentation GUI algorithm (Bintu *et al*, 2016) (a gift from the Elowitz Lab at Caltech) with minor modifications. Briefly, fluorescence images were first background subtracted, and cells were segmented using the nuclear H2B signals. Tracking was automatically performed based on the segmented cell nuclei. The cell masks and the tracking information were then used to extract the nuclear fluorescence of the constitutive FP as well as the inducible FP. The tracked cells were manually inspected in the GUI to remove tracks with dead cells or segmentation/tracking errors. After these procedures, mean nuclear fluorescence trajectories were then exported and used for further calculations such as fitting.

### Obtaining the relative intensity ratio between inducible and constitutive FPs

As required by the kinetic models, we needed to achieve relative quantification of the two FPs, i.e., the inducible FP and the constitutive FP. That is, the fluorescence level from one FP needs to be scaled with the fluorescence level from the other FP for model fitting purposes.

Thus, for each cell line, we performed separate microscopy measurements to quantify the scaling ratio between the two

fluorescence signals (i.e., the parameter R in our model). The experimental time course was illustrated in the lower panel of Fig EV1C. More specifically, in such experiments, we first added 0.75 μg/ml doxycycline to the culture at time zero, after which images were acquired continuously for 30 min with a frame rate of 1 per 3 min. These data were used for quantifying the background fluorescence. After another 10 min, 2 μg/ml actinomycin D (Coolaber) was added to the culture to block RNA synthesis in order to allow cells to reach steady fluorescence levels. Reaching steady-state fluorescence levels is important for accurate calculation of the scaling ratios. To experimentally determine this steady state (and without introducing too much phototoxicity), we started another round of image acquisition 7 h after actinomycin addition for 2 h, with a frame rate of 1 per 20 min. These data allowed us to determine whether and when steady-state fluorescence levels had been reached, and were used for calculating the intensity scaling ratio between the inducible FP and the constitutive FP.

To calculate the scaling ratio (i.e., the parameter R in the model), the nuclear levels of the two FPs were first calculated by subtracting the steady-state levels (determined from the second movie above) with the background fluorescence levels (determined from the first movie above). The scaling ratio was then calculated for each pair of the inducible FP and the constitutive FP and was used in the model fitting.

**Data fitting and maturation time calculation**

The kinetic model that characterizes inducible FP (FP2 in Fig 2A) expression and maturation is shown below.

$$DNA \xrightarrow{k_1} RNA\ (m) \xrightarrow{k_2} \underset{FP\ (I)}{Unmatured} \xrightarrow{k_3} \underset{FP\ (M)}{Matured}$$
$$\alpha \downarrow$$
$$\varnothing$$

In this model, mRNA, noted by the variable $m$, is constantly produced at a rate of $k_1$ after doxycycline addition. The mRNA has a first-order degradation rate constant of $\alpha$ and is translated into unmatured FP (variable $I$) at a rate of $k_2$. The maturation of FP includes chromophore folding and subsequent maturation and is approximated as a first-order reaction with a rate constant $k_3$. The maturation time is then defined as $ln2$ divided by $k_3$. It should be noted that we omitted the degradation at the protein level due to the short experimental durations.

In the two-color assay (Fig 2A), we used the nuclear level of the constitutive FP (FP1 in Fig 2A) to measure the protein level of the inducible FP (FP2, the sum of matured and unmatured) and used the fluorescence of FP2 to measure its own maturation. Therefore, the reactions in the model can be expressed as:

$$\frac{dm}{dt} = k_1 - am \tag{1}$$

$$\frac{d(F_c R)}{dt} = k_2 m \tag{2}$$

$$I = F_c R - F_i \tag{3}$$

$$\frac{d(F_i)}{dt} = k_3 I \tag{4}$$

Here, $m$ denotes the mRNA level, $I$ denotes the level of unmatured FP2, $k_1$ is the mRNA production rate, $\alpha$ is the mRNA turnover rate, $k_2$ is the translation rate, $k_3$ is the FP2 maturation rate, $F_i$ is the nuclear fluorescence level of FP2, $R$ is the scaling ratio, and $F_c$ is the nuclear fluorescence level of FP1.

To solve these simple ODEs, we first set the mRNA turnover rate $\alpha$ to be 0.03/min for simplicity, which is chosen based on previous estimations (Bintu *et al*, 2016). Next, the equations can be easily solved in an analytic manner to obtain the expressions for $F_c$ and $F_i$, the experimentally measured fluorescence levels. The two expressions contain only one $(k)$ and two $(k$ and $k_3)$ effective parameters, respectively, where $k = k_1 \cdot k_2$.

These expressions were then fitted to the measured fluorescence levels using standard non-linear fitting tools in MATLAB, and the maturation rate constant $k_3$ was obtained on a cell-by-cell basis. To ensure good fitting quality, we only kept cells with $R^2$ larger than 0.98. The FP maturation time of the remaining cells was then calculated. However, because there are occasionally cells with abnormal maturation kinetics such as dying cells, we removed outliers by using the median absolution deviation approach (Leys *et al*, 2013). The resulting cells were used to calculate 95% confidence intervals using bootstrap. More specifically, the data were resampled 2,000 times with replacement, and the bias-corrected 95% confidence interval of those 2,000 samples was represented as error bars in Fig 3A.

**Comparing nuclear versus cytoplasmic FP maturation kinetics**

A plasmid containing two identical FPs that are driven by a bidirectional tetracycline-inducible promoter (pTRE-BI) was co-transfected with an H2B labeling FP (with a different color) into CHO cells. Monoclonal cells were isolated by FACS as described in the earlier section. Cells were seeded in a 24-well glass-bottom plate, and fluorescence images were taken on a high-content microscope (Molecular Devices). Culture media containing 0.1 μg/ml doxycycline were added at time zero, and the frame rate was 1 per 30 min. The resulting images were processed in order to extract the nuclear versus cytoplasmic fluorescence signals. The nuclear region is defined by the H2B signal while the cytoplasmic region is defined as a 4-pixel-wide ring surrounding the nucleus. The ratio of nuclear versus cytoplasmic fluorescence signals was calculated by the mean nuclear fluorescence level divided by the median cytoplasmic fluorescence level.

**Characterizing the effect of cell cycle on maturation kinetics**

We used cell cycle synchronization to explore the effect of cell cycle on FP maturation. To avoid imaging mitotic cells, whose images are difficult to analyze, we synchronized CHO cells to G1 and S phases, and released cells at different time points after synchronization in order to study the effect of different cell cycle stages (Ma & Poon, 2017). More specifically, synchronization to G1 phase was achieved with lovastatin which inhibits HMG-CoA reductase, and synchronization to S phase was achieved with a double thymidine block procedure. After synchronization, cells were washed twice with DPBS to release cells from cell cycle blockages.

During the experiments, we induced the protein expression at different time points post-release (e.g., 0 and 2 h) and classified

cells based on their initial nuclear sizes. Such classification allowed us to analyze whether cells that started the protein induction at different cell cycle stages have different maturation kinetics. One-way analysis of variance (ANOVA) was performed to compare the maturation rates in three separate cell populations starting at different cell cycle stages.

### Stochastic simulations of FP maturation

We hypothesized an underlying noisy biochemical environment affecting the FP maturation reaction. To implement how such noisy environment affects FP maturation in the model simulation, we used a stochastic time series to represent the noisy maturation rate constant $k$. This stochastic time series has a fixed time lag that is much shorter than the maturation time, but has fluctuating amplitudes that are drawn from a Gaussian distribution centered around the mean rate constant (defined by the FP maturation time). This noisy time series of rate constant was then used for stochastic simulation of FP maturation. Note that this stochastic time series was constructed independently for each run of simulation (i.e., for each *in silico* cell). During the simulation, we implemented the standard Gillespie algorithm (Gillespie, 1977), which specifies the dwell time between two reaction steps. We averaged the time series above over the specified dwell time to obtain the maturation rate constant (i.e., time-averaging) and used it for each Gillespie simulation step. Thus, the noisy biochemical environment was effectively averaged over a time window that relates to the length of FP maturation time. When comparing two FPs of different maturation times, the faster-maturing FP would have a smaller time window for averaging and its maturation rate would be effectively more susceptible to the noisy biochemical environment compared to the slower-maturing FP.

For each FP with a chosen maturation time, by repeating the simulations of the maturation of 2,000 FP molecules in 100 cells, we generated a distribution of FP maturation time (i.e., the time when half of the molecules mature). These data were then used to calculate the mean maturation time and the noise of maturation time by bootstrap (Fig 4C).

### Characterizing maturation kinetics in media with reduced oxygen levels

EC-Oxyrase stock solution (Sigma #SAE0010, 30 units/ml) was diluted (0–3 μl oxyrase solution in 200 μl final volume, i.e., 0:200, 1:200, 2:200, or 3:200) in culture media supplemented with 10 mM DL-lactate (Sigma), which serves as the substrate for the oxyrase enzyme. The media were mixed and kept at 37°C for 2 h to ensure sufficient removal of oxygen. The resulting media were added into cells, and the expression of inducible FP was induced at the same time. The same experimental and data analysis procedures as above were then applied for these cells, except that a frame rate of 1 per 10 min was used instead of 1 per 6 min.

## Data availability

The dataset and computer code produced in this study are available in the following databases:

- Fluorescence filter information for the fluorescent proteins used in this study is available in Dataset EV1.
- Nucleotide sequences for the fluorescent proteins used in this study are available in Dataset EV2.
- Computer code for stochastic simulation is available in Code EV1.

**Expanded View** for this article is available online.

## Acknowledgements

This work was supported by grants from National Natural Science Foundation of China (Grant No. 31771425) and Ministry of Science and Technology of China (Grant No. 2018YFA0900703). We thank the flow cytometry core at the National Center for Protein Sciences and the imaging facility at the Center for Quantitative Biology at Peking University.

## Author contributions

JW and YL conceptualized the experiments and data analysis. JW performed the experiments and data analysis. XH, HZ, and TY contributed to the experiments and data analysis. JW and YL wrote and edited the paper.

## Conflict of interest

The authors declare that they have no conflict of interest.

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
