## [Review Process File · Molecular Systems Biology]

Evidence for rate-dependent filtering of global extrinsic noise by biochemical reactions in mammalian cells

Jiegen Wu, Xu Han, Haotian Zhai, Tingyu Yang, and Yihan Lin,

Review timeline:	Submission date:	4 th December 2019
	Editorial Decision:	7 th December 2019
	Revision received:	14 th March 2020
	Editorial Decision:	30 th March 2020
	Revision received:	31 st March 2020
	Accepted:	3 rd April 2020

Editor: Jingyi Hou

Transaction Report:

1st Editorial Decision

7th December 2019

Thank you for submitting your work to Molecular Systems Biology. We have now heard back from the three reviewers who agreed to evaluate your manuscript. You will see from the comments below that the reviewers find the manuscript to be of interest. They raise however a series of concerns, which should be convincingly addressed in a major revision.

Without reiterating all the points raised in the reviews below, some of the more substantial issues are the following:

- Reviewer #1 raises a series of conceptual concerns regarding noise, which need to be addressed.
- Reviewer #2's concerns regarding the experiments for perturbing the extrinsic noise source should be convincingly addressed.

All other issues raised by the reviewers need to be satisfactorily addressed as well. During our pre-decision cross-commenting process (in which the reviewers are given the chance to make additional comments, including on each other's reports), Reviewer #1 and Reviewer #2 acknowledge that while the FP maturation rate has been quantified before (as commented by Reviewer #3), they agree that it is not the main point of the current study.

REFeree REPORTS

Reviewer #1:

This manuscript aims to investigate the effects of global extrinsic noise through fluorescence measurements indicating the maturation of fluorescent reporters. The method is based on SynZip domains, which zip two reporters together. The first reporter is constitutively expressed and cytoplasmic. As the second NLS-tagged reporter is expressed, it zips to the first reporter and pulls it

into the nucleus, a process that reflects the levels of reporter 2. As the 2nd fluorophore matures, fluorescence at a new wavelength reports the maturation of reporter 2. Measuring 14 reporters reveals a variety of maturation times, which correlate inversely with the coefficient of variation (CV) of protein maturation. On the other hand, the correlation with the CV of protein production is non-significant. NADPH and oxygen levels alter noise as expected based on effects on maturation time.

Overall, this is an interesting and well-written paper that is interesting to a broad audience. I would like to suggest publication once the following comments can be addressed.

- (1) There are some conceptual issues with statements in the manuscript that need to be addressed. For example, "for a typical cellular reaction, the reactants and products have very low copy numbers" is probably not true. Whereas DNA and some mRNA species are present in low copy numbers, the same is not true for most cellular reactants, including metabolites, enzymes, small molecules. So "typical" is not the right word to use.
- (2) Figure 1B: The cartoon illustration of degradation is conceptually incorrect. The line in the figure wiggles up and down, which is impossible in a degradation process. In a degradation process, concentrations change only downwards (they drop but never increase). In fact, Fig. 1B could be redone using simple stochastic simulations.
- (3) The result that slower-maturing reporters have less noise in their maturation time is somewhat expected if they integrate incoming noise during their maturation. This is like a running average - the longer the averaging window, the lower the noise. The noise's dependence on maturation time indicates that some common underlying noisy process affects all reporters. What is that process?
- (4) While the noise of protein production does not correlate with maturation, what about the degradation? Some proteins degrade as they mature, which affects the observed maturation times. Can somehow the relationship between degradation and maturation be investigated, experimentally or computationally, or can this be at least discussed?
- (5) Fig. 2D: it would help the reader if the symbols would represent the emission color of the reporter. So iRFP should not be blue. It should be red. Distinguish the reporters by different symbols, make their colors correspond approximately to their emission wavelength. The same applies to Figure 3 and all other figures where various reporters are represented.
- (6) How can it be claimed that all maturation noise is extrinsic, originating from upstream processes? This is unlikely to be true. If random reactions happen, there should be some intrinsic noise as well. One way to demonstrate it would be to measure in the same cell the maturation time variability for two reporters with similar means, like mCherry and mClover3, in a way analogous to PMID:12237400.
- (7) Figures 2 and 3 collapse information too much. It would be better to see the distributions of maturation times (not just their means and CVs). Are these distributions similar?
- (8) The model is problematic. What does it mean to "impose white noise to the first-order maturation rate constant"? White noise is defined as an uncorrelated process in time, but "white" in simulations may reflect the time steps of simulation. It may be important to alter the time step and repeat the simulations. Was the rate constant an uncorrelated temporal process then? Or were rate values pulled randomly from some distribution each time and the simulations run repeatedly?
- (9) How would the results change for other proteins (not fluorescent reporters)? What steps in the formation of other proteins - folding (?) be in common with the reporters? What general conclusion can be drawn from the reporters about other proteins in general?
- (10) Regarding "non-genetic cell-to-cell variability in physiological states, including cell growth and drug resistance", some useful recent references would be PMID:31754027, PMID:31235692 and PMID:30341217, which would be worth citing.

Reviewer #2:

In this manuscript, the authors present an analysis of the effects of extrinsic noise on fluorophore maturation rates in mammalian cells. They describe a novel system to decouple protein production rate measurements from maturation rate measurements using the nuclear translocation of heterodimerized fluorophores. Using the method, they quantify the maturation rates in live cells for a wide range of fluorescent proteins. They find that maturation rates are inversely proportional to the coefficient of variance in the measurement; in contrast, production rates have uniform noise levels. They show that an inverse relationship between maturation rate and extrinsic noise can be described by a straightforward stochastic model of fluorescent protein production and fluorophore maturation. They use two methods to experimentally test the relationship, scavenging oxygen in the cell growth

medium and testing different cell line backgrounds.

Overall, this is an interesting method to monitor fluorescent protein production and maturation rates in single living cells. Characterization of the effects of extrinsic noise on these processes may be of potential importance for a broad range of biological questions regarding noise in regulating cellular processes. Listed below are several suggestions for how this manuscript may be improved.

Major comments:

1. In the measurements of maturation rates using the novel system, it would be good to include a negative control verifying that doxycycline does not affect the production or localization of the constitutive FP1. There is no reason to think dox would have an effect, so this should be a straightforward control.
2. On page 7, the authors state that they have shown that "the level of noise depends on the rate of maturation reaction" and "the noise in maturation time increases as the maturation time shortens." This is incorrect, they have only shown, especially at this point in the manuscript, that there is an inverse correlation between the two quantities. The statements should be weakened for improved accuracy.
3. The shaded error indications in the time traces in Fig S4B are difficult to interpret. Color coded error bars may be helpful. Additionally, the authors' claims would be strengthened if they performed a statistical test showing that the FP production values are not significantly different but that the FP maturation rates are significantly different in the experiment shown in S4B.
4. The experiments using oxyrase to alter medium oxygen levels are not particularly convincing. There appears to be an increase in noise with the 1:200 dilution, no change with the 2:200 dilution, and a decrease with the 3:200 dilution compared with the 0:200 negative control. The authors claim that the noise decreases with increasing oxyrase concentration, but that is only true for the one experimental condition of the highest concentration. What is also worrying about relying on just that one condition is that the number of cells analyzed for that condition was much lower than for the other three conditions (16 cells vs 120, 125, and 52 cells). To support the claim better, a wider range of doses should be analyzed with a greater number of cells analyzed for each dose. Additional appropriate statistical analysis to support the claim should also be provided.
5. For the experiment in Fig 5D, using different cell types is not a particularly clean experimental comparison. The authors provide a reasonable hypothesis based on variability in biliverdin levels as a method to alter maturation rates. I would suggest altering biliverdin levels within the same cell type is a more precise experiment. One method might be changing the expression of biliverdin reductase through an inducible expression system. As an additional comment on the original experiment presented in Fig 5D, again there are low cell numbers analyzed (11, 42, 14, and 19 cells).

Minor comments:

1. In Fig S3A and S3B, the correlations of interest are those with the maturation time. Many other correlations are quantified, so I would suggest drawing a box to better highlight the relevant comparisons.
2. On page 6, there is an incorrect figure reference: "the cell cycle for the FP maturation reactions (Fig. S2D)" should actually reference Fig S3D.
3. The color coding of the different fluorophores between Figs 2D, 3A, and 3B are inconsistent and non-intuitive. At the very least, the same color should be used consistently for each fluorophore throughout the manuscript. Ideally, a color closer to the actual emission color would be logical.
4. The figure legend for the experiments with oxyrase (Fig 5B and Fig S4B-E) are not clear. What do the dilution ratios mean in terms of actual enzyme concentration? It is not sufficiently described in the Results, Methods, or Figure Captions.
5. On page 11, something is missing in the phrase "influence other cellular processes such as and cell growth."
6. On page 16, "Rsquared" should be corrected.

Reviewer #3:

Wu et al devised a system of dimerizing fluorescent proteins with different maturation times. The nuclear translation of the dimerized proteins enabled them to measure the maturation time of the

fluorescent proteins. Furthermore, they found an inverse relation between maturation time and gene expression noise, which is due to the time averaging.

Fluorescent proteins of different maturation times have been used as timers, which permits for example, to infer the rates of protein degradation (and translocation) "without the need for time course measurements (A. Khmelinskii et al., Nat. Biotech., 30:708, 2012)."

<http://book.bionumbers.org/what-is-the-maturation-time-for-fluorescent-proteins/>

In the above source, we can also see the maturation times of the fluorescent proteins. While the authors correctly state that most of the measurements have been made with bacteria and yeast, their results with the mammalian cells yield similar data and confirm that red fluorescent proteins have typically slower maturation than green one.

While the design is interesting, most of the findings are relatively well-known or expected. It is possible, that the full potential of the system has not been yet tapped. However, it is up to the authors to come up with a more convincing application that takes advantage of the designed system.

1st Revision - authors' response

14th March 2020

Point by point reply:

Reviewer #1:

This manuscript aims to investigate the effects of global extrinsic noise through fluorescence measurements indicating the maturation of fluorescent reporters. The method is based on SynZip domains, which zip two reporters together. The first reporter is constitutively expressed and cytoplasmic. As the second NLS-tagged reporter is expressed, it zips to the first reporter and pulls it into the nucleus, a process that reflects the levels of reporter 2. As the 2nd fluorophore matures, fluorescence at a new wavelength reports the maturation of reporter 2. Measuring 14 reporters reveals a variety of maturation times, which correlate inversely with the coefficient of variation (CV) of protein maturation. On the other hand, the correlation with the CV of protein production is non-significant. NADPH and oxygen levels alter noise as expected based on effects on maturation time. Overall, this is an interesting and well-written paper that is interesting to a broad audience. I would like to suggest publication once the following comments can be addressed.

We thank the reviewer for the very helpful comments. In response to these comments, we have performed new experiments and simulations, which have greatly strengthened the conclusions in the manuscript.

(1) There are some conceptual issues with statements in the manuscript that need to be addressed. For example, "for a typical cellular reaction, the reactants and products have very low copy numbers" is probably not true. Whereas DNA and some mRNA species are present in low copy numbers, the same is not true for most cellular reactants, including metabolites, enzymes, small molecules. So "typical" is not the right word to use.

We thank the reviewer for pointing this imprecise expression. The rewritten sentence (**Line 33-34**) now reads, "A key reason is that for some cellular reactions, the molecular species involved often have low copy numbers and ...". In addition,

we made several minor edits in other parts of the text to ensure accuracy, for example, **Line 39**, **Line 51**, and several other locations.

(2) Figure 1B: The cartoon illustration of degradation is conceptually incorrect. The line in the figure wiggles up and down, which is impossible in a degradation process. In a degradation process, concentrations change only downwards (they drop but never increase). In fact, Fig. 1B could be redone using simple stochastic simulations.

The reviewer raised a very good point - the cartoon is indeed inaccurate. Per the reviewer's suggestion, we have now replaced the original sketch in **Fig. 1B** with a simulated trace.

(3) The result that slower-maturing reporters have less noise in their maturation time is somewhat expected if they integrate incoming noise during their maturation. This is like a running average - the longer the averaging window, the lower the noise. The noise's dependence on maturation time indicates that some common underlying noisy process affects all reporters. What is that process?

We are glad that based on the data the reviewer reached the same hypothesis as we proposed in the manuscript, i.e., there is an underlying noisy process affecting the maturation of all fluorescent protein reporters. And indeed, the measured dependence of noise level on maturation rate can be explained by a time-averaging-based noise filtering mechanism.

As for the potential source of the underlying noise, we speculated that it could be fluctuations in the redox environment affecting the oxidation of the FPs. This speculation is based on the following reasons: 1) the oxidation reaction during the maturation process is the rate-limiting step and is subjected to the cellular redox environment such as the NADPH level; 2) our results showed that FP maturation rate is robust to diverse cell-state parameters but is highly susceptible to environmental oxygen level. While our results indicate that such noise globally affects biochemical reactions, the nature of the underlying mechanism remains undetermined due to the lack of fast-responsive metabolic reporters.

In order to make these points more clearly in the manuscript, we have expanded the discussion on the potential source of noise in the main text (see **Line 347-351**).

(4) While the noise of protein production does not correlate with maturation, what about the degradation? Some proteins degrade as they mature, which affects the observed maturation times. Can somehow the relationship between degradation and maturation be investigated, experimentally or computationally, or can this be at least discussed?

The reviewer raised a good point regarding the potential influence by protein degradation. For our experiments, we think that such influence should be very small because of the following reason. The FP reporters used in our experiments are stable as there are no destabilization signals fused to these reporters. Reporter degradation can thus only occur passively through the dilution by cell division, which is much slower compared to FP maturation. And during image analysis, we only analyzed cells that did not undergo mitosis during the time course of the experiment. Therefore, we believe that protein degradation should have minor contribution to the measured fluorescence signals.

We next used stochastic simulations to investigate the potential role of protein degradation on FP maturation noise. It should be noted that the simulation implements a constant degradation rate with a half-life of 20 hrs (i.e., cell cycle time), which does not necessarily reflect the actual process of passive degradation discussed above. Gillespie algorithm was used for stochastic simulations. The simulated results show that the presence of protein degradation does not alter the noise level of the FP maturation time in a statistically significant manner (**Figure R1**).

To reflect these findings, we now include a discussion on the potential effect of protein degradation (see **Line 201-204**).

Figure R1. *In silico* study of the potential role of protein degradation on the noise in maturation reaction. Stochastic simulations were performed to investigate the effect of protein degradation for the maturation of FPs with different maturation times. For protein degradation, a half-life of 20 hrs was used. Each condition contains 100 cells in the simulation and each cell contains 2000 molecules. Error bars indicate 95% confidence intervals of the mean by bootstrap.

(5) Fig. 2D: it would help the reader if the symbols would represent the emission color of the reporter. So iRFP should not be blue. It should be red. Distinguish the reporters by different symbols, make their colors correspond approximately to their emission wavelength. The same applies to Figure 3 and all other figures where various reporters are represented.

We thank the reviewer for the suggestion. We have re-drawn **Fig. 2D** and **Fig. 3** to ensure that the colors of the FP data symbols are in close correspondence to

their emission wavelength. We have also made minor changes in the colors for several other figures to enhance readability.

(6) How can it be claimed that all maturation noise is extrinsic, originating from upstream processes? This is unlikely to be true. If random reactions happen, there should be some intrinsic noise as well. One way to demonstrate it would be to measure in the same cell the maturation time variability for two reporters with similar means, like mCherry and mClover3, in a way analogous to PMID:12237400.

We thank the reviewer for raising an important point regarding the source of noise. As suggested by the reviewer, we used a pseudo “two-color” assay (analogous to the assay used by Elowitz et al, 2000) to analyze the relative contributions of extrinsic versus intrinsic noise. The result supports our conclusion in the manuscript that the extrinsic noise should be the major source of the measured noise. Below we describe the details regarding this pseudo “two-color” assay. Note that the term “two-color” is just to draw an analogy between our assay and the original two-color assay.

We first explain the rationale for our “two-color” assay. Because our method to measure maturation requires two FPs for each FP of interest, it is thus extremely challenging to measure the maturation of two FPs of interest within the same cell. We thus turned to an alternative assay, where two proteins containing the same FP but different localization signals (i.e., nuclear versus cytoplasmic) are co-expressed in the same cell under the control of a bidirectional promoter. Due to the differential localization signals, fluorescence from the two proteins can be spatially separated and independently quantified. The same cell line was used to ensure that FP maturation is unaffected by its nuclear or cytoplasmic localization (**Fig. EV2A**).

We performed the analysis of extrinsic and intrinsic noises (as described by Elowitz et al, 2000) after the addition of inducer doxycycline (**Fig. EV2D**). We found that the extrinsic noise is the dominant source of noise for the FP fluorescence. And because the measured intrinsic noise level is much smaller than the noise level measured from the maturation assay (**Fig. EV2D** vs. **Fig. 3A**), we reason that the noise in the maturation reaction is largely extrinsic. This conclusion is consistent with the results from our model simulation (**Fig. 4D**), where increasing the influence from intrinsic fluctuations (i.e., by decreasing the number of molecules) did not increase the overall noise level.

To clarify this point in the manuscript, we added a panel in **Fig. EV2** and included a new paragraph in the text (see **Line 206-212**).

(7) Figures 2 and 3 collapse information too much. It would be better to see the distributions of maturation times (not just their means and CVs). Are these distributions similar?

We thank the reviewer for suggesting means to improve data representation. We have replotted **Fig. 2** using boxplot to better present the data. We think that

boxplot provides a better visualization for the data compared to the original representation.

To compare the distributions of FP maturation times, we plotted the normalized distributions below in **Figure R2**. By comparing the distributions of all measured FPs, we found that most of the distributions are similarly shaped and the widths of the normalized distributions vary across FPs in a manner that agrees with trend in the measured noise level.

Figure R2. Distributions of maturation times for 14 FPs. In order to compare between different FPs, each distribution was normalized by its mean value such that it is centered around one.

(8) The model is problematic. What does it mean to "impose white noise to the first-order maturation rate constant"? White noise is defined as an uncorrelated process in time, but "white" in simulations may reflect the time steps of simulation. It may be important to alter the time step and repeat the simulations. Was the rate constant an uncorrelated temporal process then? Or were rate values pulled randomly from some distribution each time and the simulations run repeatedly?

We realize that the original model description was not clear enough. We apologize for the confusion caused by the original descriptions.

Here we try to clarify the model, which would hopefully address the questions raised by the reviewer. As discussed above (point 3), we hypothesized an underlying noisy biochemical environment affecting the FP maturation reaction. To implement how such noisy environment affects FP maturation in the model simulation, we used a stochastic time series to represent the noisy maturation rate constant k . This stochastic time series has a predefined fixed time lag that is much shorter than the maturation time, but has fluctuating amplitudes that are drawn from a Gaussian distribution centered around the mean rate constant (defined by the FP maturation time). This noisy time-series of rate constant was then used for stochastic simulation of FP maturation. Note that this stochastic time-series was constructed independently for each run of simulation (i.e., for each *in silico* cell). During the simulation, we implemented the standard Gillespie algorithm, which specifies the dwell time between two reaction steps. We averaged the time series above over the specified dwell time to obtain the maturation rate constant (i.e., time-averaging) and used it for each Gillespie simulation step. Thus, the noisy biochemical environment was effectively averaged over a time window that relates to the length of FP maturation time. When comparing two FPs of different maturation times, the faster-maturing FP

would have a smaller time window for averaging and its maturation rate would be effectively more susceptible to the noisy biochemical environment compared to the slower-maturing FP.

To better clarify the model, we have now greatly expanded the description of the model in the Methods section (see **Line 535-550**).

(9) How would the results change for other proteins (not fluorescent reporters)? What steps in the formation of other proteins - folding (?) be in common with the reporters? What general conclusion can be drawn from the reporters about other proteins in general?

We thank the reviewer for raising the question regarding the generality of our findings.

First, the chromophore oxidation step (instead of protein folding) is the rate-limiting step during the maturation of FPs according to the literature, and the maturation reaction is often modeled as a first-order chemical reaction. As such, the oxidation step is most susceptible to fluctuations in biochemical environment, which is supported by the susceptibility of FP maturation to oxygen level, as demonstrated in our data (and in others' as well). Thus, in the manuscript, we hypothesized that FP maturation reaction is subjected to temporal concentration fluctuations in metabolites (such as NADPH) related to cellular redox environment.

Based on these results from FPs, we next address what general conclusions could be drawn for non-FP proteins. The activities of many enzymes such as biosynthetic enzymes are dependent on the cellular redox environment and are likely subjected to stochastic fluctuations of this environment (such as fluctuating NADPH level), similar to the scenario for FPs. It is thus tempting to speculate that key biosynthetic pathways could be affected by global extrinsic noise and may thus confer heterogeneity in the pathway output across isogenic cells. Similar speculations could be made regarding the redox signaling pathways and related metabolic pathways. Therefore, the presence of global extrinsic noise could potentially influence a broad range of enzymes and pathways and contribute to non-genetic cell-to-cell variability.

To reflect this point, we have now expanded the discussion on the generality of our conclusions (see **Line 347-351**)

(10) Regarding "non-genetic cell-to-cell variability in physiological states, including cell growth and drug resistance", some useful recent references would be PMID:31754027, PMID:31235692 and PMID:30341217, which would be worth citing.

We thank the reviewer for suggesting these interesting papers that are relevant to our work. We have now included these citations in the introduction section (**Line 53**).

Reviewer #2:

In this manuscript, the authors present an analysis of the effects of extrinsic noise on fluorophore maturation rates in mammalian cells. They describe a novel system to decouple protein production rate measurements from maturation rate measurements using the nuclear translocation of heterodimerized fluorophores. Using the method, they quantify the maturation rates in live cells for a wide range of fluorescent proteins. They find that maturation rates are inversely proportional to the coefficient of variance in the measurement; in contrast, production rates have uniform noise levels. They show that an inverse relationship between maturation rate and extrinsic noise can be described by a straightforward stochastic model of fluorescent protein production and fluorophore maturation. They use two methods to experimentally test the relationship, scavenging oxygen in the cell growth medium and testing different cell line backgrounds.

Overall, this is an interesting method to monitor fluorescent protein production and maturation rates in single living cells. Characterization of the effects of extrinsic noise on these processes may be of potential importance for a broad range of biological questions regarding noise in regulating cellular processes. Listed below are several suggestions for how this manuscript may be improved.

We thank the reviewer for the very helpful comments. In response to these comments, we have performed several new experiments and analysis, which have greatly strengthened the conclusions in the manuscript.

Major comments:

1. In the measurements of maturation rates using the novel system, it would be good to include a negative control verifying that doxycycline does not affect the production or localization of the constitutive FP1. There is no reason to think dox would have an effect, so this should be a straightforward control.

We thank the reviewer for suggesting this important control experiment. We have performed a new experiment showing that doxycycline indeed does not affect the production or localization of the constitutive FP1.

More specifically, we carried out an assay in CHO cells to test whether doxycycline affects the production or localization of the constitutive FP1. We transiently transfected plasmids containing the constitutively expressed mTurquoise2 (FP1) and the plasmids containing the constitutively expressed nuclear labeling iRFP (iRFP-H2B) into CHO cells. Time-lapse microscopy (with a frame rate of 1 per 15 min) was performed on this cell line for 135 minutes. During the first hour of the experiment, the culture medium did not contain doxycycline, after which the medium was replaced with new medium containing doxycycline (1 $\mu\text{g}/\text{mL}$).

We next compared the fluorescence signals before and after doxycycline addition. To quantify the potential changes in the nuclear localization or expression level, we performed cell segmentation and cell tracking on the

fluorescence images, and acquired the nuclear fluorescence trajectories of both mTurquoise2 and iRFP. Each trajectory was normalized by its mean fluorescence. We plotted the mean and standard deviation of normalized fluorescence in **Fig. EV1D**. By comparing the data from pre- and post- dox addition, we found that both fluorescence signals are centered around the mean value (i.e., one), suggesting that doxycycline does not affect the production or localization of the constitutive FP1.

To incorporate these results, we have included a sentence in the main text (**Line 123-124**) and a new panel in **Fig. EV1**.

2. On page 7, the authors state that they have shown that "the level of noise depends on the rate of maturation reaction" and "the noise in maturation time increases as the maturation time shortens." This is incorrect, they have only shown, especially at this point in the manuscript, that there is an inverse correlation between the two quantities. The statements should be weakened for improved accuracy.

We thank the reviewer for pointing out this inaccurate statement. Indeed, we have not yet reached a causal relationship between noise level and maturation rate at this point of the text. We have replaced the expression by emphasizing the correlation that we observed. The new sentence now reads "*the level of noise exhibits a correlation with the rate of maturation reaction*" (see **Line 191-192**).

3. The shaded error indications in the time traces in Fig S4B are difficult to interpret. Color coded error bars may be helpful. Additionally, the authors' claims would be strengthened if they performed a statistical test showing that the FP production values are not significantly different but that the FP maturation rates are significantly different in the experiment shown in S4B.

We thank the reviewer for suggesting approaches to improve data representation and to strengthen statistical analysis of the data.

Per the reviewer's suggestion and to allow better interpretation and comparison, we have now replaced the original plot with new plots showing the mean and error bars (s. d.) of six time points for both FP1 and FP2 signals under all oxyrase conditions (**Fig. EV4B**). We then performed multiple comparison tests to determine whether oxyrase concentration significantly affected either protein production (i.e., using FP1 signals) or FP maturation (i.e., using FP2 signals). We found that there are no significant differences among FP1 signals under all oxygen conditions while the FP2 signals are significantly different ($p < 0.05$). These results suggest that the FP maturation rate but not the FP production rate was altered by changes in oxygen concentration.

Thus, our original conclusion was greatly strengthened by this new analysis. In addition to **Fig. EV4B**, we have included a similar analysis for the experiments with biliverdin (**Fig. EV5A-B**), showing the protein production is similarly unaffected by the addition by biliverdin.

4. The experiments using oxyrase to alter medium oxygen levels are not

particularly convincing. There appears to be an increase in noise with the 1:200 dilution, no change with the 2:200 dilution, and a decrease with the 3:200 dilution compared with the 0:200 negative control. The authors claim that the noise decreases with increasing oxyrase concentration, but that is only true for the one experimental condition of the highest concentration. What is also worrying about relying on just that one condition is that the number of cells analyzed for that condition was much lower than for the other three conditions (16 cells vs 120, 125, and 52 cells). To support the claim better, a wider range of doses should be analyzed with a greater number of cells analyzed for each dose. Additional appropriate statistical analysis to support the claim should also be provided.

We acknowledge the issue raised by the reviewer regarding the oxyrase experiments. As the reviewer pointed out, the conclusion was not convincingly supported by the original data set. We have overcome technical difficulties and acquired a much larger data set in order to gain more statistical power. The key technical difficulty was the increased photo-toxicity at high oxyrase concentration, prohibiting us from obtaining a large number of cells. We have thus revised the protocol for image acquisition by reducing the frame rate from 1 per 6 min to 1 per 10 min. With this new protocol, we were able to reduce photo-toxicity and obtain much more cells compared to the original data set. This new set of data contains 143, 395, 299, and 529 cells for the four oxyrase dilution conditions (0:200, 1:200, 2:200, and 3:200), respectively. Note that we did not perform the assay at higher oxyrase levels due to the technical challenge above.

Figure 5B has now been updated with the analysis of these new data. We think that this new result provides a strong support for the claim that the noise level decreases when increasing the oxyrase concentration. We have also modified the method section to reflect the changes (see **Line 564-565**)

5. For the experiment in Fig 5D, using different cell types is not a particularly clean experimental comparison. The authors provide a reasonable hypothesis based on variability in biliverdin levels as a method to alter maturation rates. I would suggest altering biliverdin levels within the same cell type is a more precise experiment. One method might be changing the expression of biliverdin reductase through an inducible expression system. As an additional comment on the original experiment presented in Fig 5D, again there are low cell numbers analyzed (11, 42, 14, and 19 cells).

We thank the reviewer for suggesting a more direct way to test the effect of cofactor biliverdin. We have performed alternative experiments for both mIFP and iRFP in the same cell type, and found that changing FP maturation rate by altering biliverdin concentration leads to changes in the noise level as the model predicted.

More specifically, instead of controlling the expression of biliverdin reductase, we chose to add biliverdin directly to the cell culture media, which has been shown to effectively increase the fluorescence of near-infrared FPs (Shemetov et al, Cell Chem Bio, 2017). We thus performed the maturation assays in monoclonal CHO cell lines for the two near-infrared FPs, mIFP and iRFP, under

two different biliverdin concentrations, i.e., 0 μM and 10 μM . Using this data set, we compared the rate of FP maturation under the two biliverdin concentrations, and found that 1) the rates of maturation for both FPs were increased by the addition of biliverdin, and 2) the noise levels were increased accordingly (**Fig. 5D**). Importantly, the protein production rates were unaffected by the addition of biliverdin. Thus, in both biliverdin and oxyrase assays, changes in biliverdin level or oxygen level affect the rate of FP maturation but not FP production (**Fig. EV5A-B**).

These findings from the biliverdin assay are consistent with our other results, where different production levels of co-factor biliverdin in CHO and U2OS cell lines contribute to differences in the maturation rates and the associated noise levels for near-infrared FPs. Therefore, results from this new experiment agree well with our model predictions and provide very strong direct support for our main conclusion. We have now replaced the original **Fig. 5D** with results from this new experiment. Note that the CHO versus U2OS results presented in the original **Fig. 5D** are now moved to the supplement (**Fig. EV6**) and used as an indirect evidence, and we think that the relatively small number of cells in this dataset is statistically sufficient to distinguish the large differences in the noise level between the two cell lines.

We have modified the text (see **Line 276-290**) and figures (**Fig. 5D, Fig. EV5-6**) accordingly to reflect these new results.

Minor comments:

1. In Fig S3A and S3B, the correlations of interest are those with the maturation time. Many other correlations are quantified, so I would suggest drawing a box to better highlight the relevant comparisons.

We thank the reviewer for the suggestion. We have highlighted the relevant features in **Fig. EV3A-3B** as suggested.

2. On page 6, there is an incorrect figure reference: "the cell cycle for the FP maturation reactions (Fig. S2D)" should actually reference Fig S3D.

We thank the reviewer for pointing out this error. We have corrected it and have also made sure that other figure references are correctly placed.

3. The color coding of the different fluorophores between Figs 2D, 3A, and 3B are inconsistent and non-intuitive. At the very least, the same color should be used consistently for each fluorophore throughout the manuscript. Ideally, a color closer to the actual emission color would be logical.

We thank the reviewer for the suggestion. We have re-drawn **Fig. 2D** and **Fig. 3** to ensure that the colors of the FP data symbols are in close correspondence to their emission wavelength. We have also made minor changes in the colors for several other figures to enhance readability.

4. The figure legend for the experiments with oxyrase (Fig 5B and Fig S4B-E) are

not clear. What do the dilution ratios mean in terms of actual enzyme concentration? It is not sufficiently described in the Results, Methods, or Figure Captions.

We thank the reviewer for pointing out this issue. We have now added detailed descriptions regarding the enzyme concentration and the method of dilution in the Methods section (see **Line 558-559**).

5. On page 11, something is missing in the phrase "influence other cellular processes such as and cell growth."

We apologize for the mistake. We have corrected the sentence (**Line 326-327**). It now reads "*influence physiological processes such as drug resistance and cell growth*".

6. On page 16, "Rsquared" should be corrected.

Thanks, we have corrected it (**Line 496**). We have also corrected several other spelling/grammar errors.

Reviewer #3:

Wu et al devised a system of dimerizing fluorescent proteins with different maturation times. The nuclear translation of the dimerized proteins enabled them to measure the maturation time of the fluorescent proteins. Furthermore, they found an inverse relation between maturation time and gene expression noise, which is due to the time averaging.

Fluorescent proteins of different maturation times have been used as timers, which permits for example, to infer the rates of protein degradation (and translocation) "without the need for time course measurements (A. Khmelinskii et al., Nat. Biotech., 30:708, 2012)."

<http://book.bionumbers.org/what-is-the-maturation-time-for-fluorescent-proteins/>

In the above source, we can also see the maturation times of the fluorescent proteins. While the authors correctly state that most of the measurements have been made with bacteria and yeast, their results with the mammalian cells yield similar data and confirm that red fluorescent proteins have typically slower maturation than green one.

While the design is interesting, most of the findings are relatively well-known or expected. It is possible, that the full potential of the system has not been yet tapped. However, it is up to the authors to come up with a more convincing application that takes advantage of the designed system.

We thank the reviewer for taking the time to comment on our manuscript. We believe that our study provides interesting findings besides what were described by the reviewer. Instead of reiterating our findings, we would like to respectfully address some of the points raised by the reviewer.

1. If we understand correctly, we are afraid that the reviewer might have misunderstood the design of the assay. Our assay to quantify maturation time was not based on the design described by the reviewer, i.e., “a system of dimerizing fluorescent proteins with different maturation times”. While we were fully aware of the timer application of tandem dimers of FPs with different maturation times, our assay did not employ an analogous concept to measure maturation time.
2. As pointed out by the reviewer, there has been a community effort to catalog the maturation times of FPs in diverse organisms, such as the awesome bionumbers project cited above as well as the FPbase website cited in our manuscript. Our systematic study of FPs in mammalian cells provides a key contribution to this effort and reveals new findings regarding the performance of FPs, such as the non-genetic heterogeneity in FP maturation (and the potential mechanism) as well as the cell line-dependent maturation speed of near-infrared FPs.
3. The reviewer noted that “the full potential of the system has not been yet tapped”. While this system has yielded scientific insights into single-cell FP maturation reaction and related cell-to-cell variability, we agreed that its full potential has not yet been unleashed, as it could also be engineered toward useful applications. As an example, we have demonstrated (in a different study) that this system could function as a multi-range timer, among other applications.

2nd Editorial Decision

30th March 2020

Thank you for sending us your revised manuscript. We have now heard back from the two reviewers who were asked to evaluate your study. As you will see the reviewers are satisfied with the modifications made and think that the study is now suitable for publication.

Before we can formally accept your manuscript, we would ask you to address a few remaining editorial issues listed below.

REFEREE REPORTS

Reviewer #1:

I would like to thank the authors for thoroughly addressing all my comments. I would like to enthusiastically recommend the publication of this highly interesting manuscript that should attract substantial attention from the quantitative biology research community and beyond. The only recommendation I have is to try adding Figure R2 (maturation time histogram) as an EV figure.

Reviewer #2:

The authors have satisfied my concerns raised regarding the initial manuscript. The manuscript has been greatly improved in terms of clarity, scientific rigor, and statistical significance of the results.

In particular, the experiment shown in Figure 5B has been improved for rigor, and the replacement experiment in Figure 5D has improved the relevance of the overall findings of this study.

2nd Revision - authors' response

31st March 2020

Reviewer #1:

I would like to thank the authors for thoroughly addressing all my comments. I would like to enthusiastically recommend the publication of this highly interesting manuscript that should attract substantial attention from the quantitative biology research community and beyond. The only recommendation I have is to try adding Figure R2 (maturation time histogram) as an EV figure.

We thank the reviewer for the time and the positive comments. We have included **Figure R2** as **Figure EV1E**, which is also properly cited in the main text (**Line 175**).

Reviewer #2:

The authors have satisfied my concerns raised regarding the initial manuscript. The manuscript has been greatly improved in terms of clarity, scientific rigor, and statistical significance of the results. In particular, the experiment shown in Figure 5B has been improved for rigor, and the replacement experiment in Figure 5D has improved the relevance of the overall findings of this study.

We thank the reviewer for the time and the positive comments.

Accepted

3rd April 2020

Thank you again for sending us your revised manuscript. We are now satisfied with the modifications made and I am pleased to inform you that your paper has been accepted for publication.

Corresponding Author Name: Yihan Lin

Manuscript Number: MSB-19-9335